# *ENO2* as a Biomarker Regulating Energy Metabolism to Promote Tumor Progression in Clear Cell Renal Cell Carcinoma

**DOI:** 10.3390/biomedicines11092499

**Published:** 2023-09-09

**Authors:** Jian Shi, Daojia Miao, Qingyang Lv, Diaoyi Tan, Zhiyong Xiong, Xiaoping Zhang

**Affiliations:** 1Department of Urology, Union Hospital, Tongji Medical College, Huazhong University of Science and Technology, Wuhan 430022, Chinamiaodj@hust.edu.cn (D.M.); d202281965@hust.edu.cn (D.T.); 2Institute of Urology, Union Hospital, Tongji Medical College, Huazhong University of Science and Technology, Wuhan 430022, China

**Keywords:** clear cell renal cell carcinoma (ccRCC), Warburg effect, glycolysis, bioinformatics, TCGA

## Abstract

Background: Clear cell renal cell carcinoma (ccRCC) is the most common and metastatic type of renal cell carcinoma. Despite significant advancements, the current diagnostic biomarkers for ccRCC lack the desired specificity and sensitivity, necessitating the identification of novel biomarkers and elucidation of their underlying mechanisms. Methods: Three gene expression profile datasets were obtained from the GEO database, and differentially expressed genes (DEGs) were screened. Gene Ontology and KEGG pathway analysis were conducted in ccRCC. To clarify the diagnosis and prognostic role of *ENO2*, Kaplan–Meier analysis and Cox proportional hazards regression analysis were performed. Functional experiments were also carried out to verify the significant role of *ENO2* in ccRCC. Finally, tumor mutational burden analysis was utilized to investigate the potential role of *ENO2* in gene mutations in ccRCC. Results: The study showed that *ENO2* is a potential biomarker for the diagnosis of ccRCC and can independently predict the clinical prognosis of ccRCC. Furthermore, we found that *ENO2* can promote the occurrence and progression of ccRCC by affecting the glycolysis level of cells through the “Warburg effect”. Conclusions: These findings provide new theories for the occurrence and development of ccRCC and can help formulate new strategies for its diagnosis and treatment.

## 1. Introduction

Renal cell carcinoma (RCC), is associated with the highest mortality rate amidst urogenital cancers, accounting for about 90% of renal cancer patients [1]. RCC is a heterogeneous epithelial tumor with multiple subtypes. As the most prevalent cancer among all histological subtypes, clear cell renal cell carcinoma (ccRCC) accounts for between 70% and 80% of renal cell carcinoma [2,3]. Although the diagnosis and treatment of ccRCC have made some progress in recent years, its incidence is still rising. At initial diagnosis, a substantial proportion of patients (nearly 30%) are metastatic, whereas 20–40% of postoperative patients develop local recurrence or metastasis [4,5]. Most patients receiving ccRCC targeted therapy worsen their condition due to acquired drug resistance [6,7]. As a result, researchers are urgently required to uncover reliable ccRCC biomarkers and molecular mechanisms, which may help early diagnosis of the disease, and to develop new treatment and prevention strategies for ccRCC.

In approximately 90% of cases of clear cell renal cell carcinoma (ccRCC), the tumor suppressor protein von Hippel–Lindau (pVHL) experiences functional loss. This results in the activation of hypoxia-inducible transcription factors, namely *HIF-2α* and *HIF-1α* [8,9]. The pseudo-hypoxic phenotype produced in ccRCC cells leads to massive angiogenesis and extensive metabolic reprogramming, including the transition from mitochondrial oxidative phosphorylation to glycolysis [10]. In contrast to normal cells, tumor cells frequently exhibit a preference for glycolytic metabolism over oxidative phosphorylation when metabolizing glucose, a phenomenon commonly known as the “Warburg effect” [11,12]. Due to the high level of activation in various tumor tissues, glycolysis status is considered to have potential prognostic value for predicting patients’ outcome [13]. In the past few decades, in vitro, in vivo, and clinical studies have used key enzymes in the glycolytic pathway as potential biomarkers for the survival of malignant tumors, such as hepatocellular carcinoma, and cancers of the lung, pancreas, gastric, breast, among others [14]. However, the role of glycolysis in ccRCC and related biomarkers have not been clarified.

Enolase 2 (ENO2) is an intracellular enzyme critically involved in the catalytic dehydration of 2-phospho-D-glycerate to phosphoenolpyruvate through the glycolytic pathway. This biochemical transformation facilitates the conversion of glucose into pyruvate, a precursor that facilitates the generation of high-energy ATP and NADH molecules [15,16]. Many enzymes of the glycolysis pathway, including enolase, play vital functions in biological processes, disease transpiration, and tumor progression [17,18,19]. *ENO2* has been proven to be a reliable biomarker in lung cancer and neuroblastoma, showing potential in a variety of tumors, such as melanoma and seminoma [20].

In this research, three independent datasets (GSE36895, GSE66272, and GSE71963) for gene expression profiling were retrieved from the Gene Expression Omnibus (GEO; https://www.ncbi.nlm.nih.gov (accessed on 10 March 2020)) to search for differentially expressed genes (DEGs) between normal kidney and ccRCC tissues. DEGs have been applied to the Kyoto Encyclopedia of Genes and Genomes (KEGG) pathway enrichment and Gene Ontology (GO) functional annotation analyses. The results showed that glycolysis exerts a critical function in ccRCC progression. We adopted the PPI (protein–protein interaction) network between DEGs to find the hub genes related to ccRCC. Upon screening and validation via the Oncomine dataset (https://www.oncomine.org (accessed on 10 March 2020)) and DEGs, *ENO2* is considered a key parameter and a promising target for ccRCC diagnosis and management. Subsequently, we substantiated the crucial role of *ENO2* within the context of ccRCC, and further explored the potential mechanisms through which it influences the progression of the disease.

## 2. Material and Methods

### 2.1. Microarray Data

Three gene expression profiling datasets (GSE36895, GSE66272, and GSE71963) were downloaded from Gene Expression Omnibus (GEO; https://www.ncbi.nlm.nih.gov/geo/ (accessed on 10 March 2020)) database. GSE36895 and GSE6627 are contingent on the Affymetrix GPL570 platform (Affymetrix Human Genome U133 Plus 2.0 Array). GSE36895 contains 76 samples, including 29 human ccRCC samples, 23 matching normal kidney tissue, and 24 mouse transplant tumor samples, of which human samples were selected. GSE66272 contains 54 samples, including 27 human ccRCC samples and 27 matched normal kidney tissues. GSE71963, contingent on the GPL6480 platform (Agilent-014850 Whole Human Genome Microarray 4 × 44K G4112F), contains 48 samples, including 32 human ccRCC samples and 16 matched normal kidney tissues.

### 2.2. Expression Analysis of DEGs

To determine those DEGs between the ccRCC tissues and matched normal kidney tissue samples in GSE36895, GSE66272, and GSE71963, we used GEO2R (an online software) when analyzing the original data of the microarray. An adjust *p* < 0.05 and log2FC > 2 were used as screening criteria. We used FunRich 3.1.4 [21] (http://www.funrich.org (accessed on 13 March 2020)) to construct a Venn diagram from the overlapping data of GSE36895, GSE66272, and GSE71963.

### 2.3. Gene Ontology and KEGG Pathway Analysis

DAVID (https://david.ncifcrf.gov/ (accessed on 20 March 2020)) is a publicly available gene function analysis bioinformation online database which was used to analyze and annotate the functions and enrichment protein pathways, which the candidate genes encode [22]. Afterward, the results produced by DAVID were visualized and analyzed with the R package “clusterprofiler” [23]. The *p*-value  <  0.05 was set as significant.

### 2.4. Protein–Protein Interactions Network and Module Analysis

Protein interaction is an important link to clarify tumor production and related molecular mechanisms. The STRING database (https://string-db.org/ (accessed on 22 March 2020)) [24] was used to establish a protein–protein interaction (PPI) network for the DEGs, which provides key integration of PPIs, including known and predicted interactions. The absolute value of the correlation coefficient score > 0.4 is used as the threshold. To visualize the PPI network, we used Cytoscape software (3.7.2) [25]. Using the same software, we adopted the Molecular Complex Detection (MCODE) plug-in to determine hub genes and high-contribution molecules in the PPI network. The relevant parameters used were as follows: degree cut-off  =  2, k-core  =  2, node score cut-off  =  0.2, and max. depth  =  100. The screened PPI network was again analyzed by functional enrichment using DAVID.

### 2.5. Data Mining in the Oncomine Database and TCGA Kidney Clear Cell Carcinoma (TCGA-KIRC)

The Oncomine database (http://www.oncomine.org (accessed on 10 March 2020)) covers most cancer types and subtype transcriptome data and unifies a large compendium of published cancer microarray data with a set of advanced analysis tools [26]. Three independent sets of glycolysis-related genes were screened from the ccRCC dataset through the Oncomine database. The expression of *ENO2* in ccRCC was verified by the Oncomine database. The clinical information of TCGA-KIRC and the expression level of *ENO2* were downloaded from XENA. We analyzed relevant data using SPSS 22.0 (IBM SPSS, Chicago, IL, USA) and generated the curves using GraphPad Prism 7.0.

### 2.6. Gene Set Enrichment Analysis

Gene set enrichment analysis (GSEA) explores the gene functions and metabolic pathways of a group of genes that are statistically significant [27]. We adopted GSEA to assess the enrichment *ENO2* pathway in the TCGA-KIRC dataset to explore the function of *ENO2* in ccRCC. *p* < 0.05, and a false discovery rate (FDR) < 0.25 denoted statistical significance.

### 2.7. Bioinformatic Analysis Using the cBioPortal for Cancer Genomics and ClueGo

#### Bioinformatic Analysis via ClueGo and the cBioPortal for Cancer Genomics

Co-expressed genes with *ENO2* in TCGA-KIRC (absolute Pearson’s r > 0.5) were detected via the cBioPortal 13.0 for Cancer Genomics (http://www.cbioportal.org/ (accessed on 20 March 2020)). Then, we imported these genes into the ClueGo app [28] in Cytoscape for KEGG pathway analysis. *p* < 0.05 denoted statistically significant differences.

### 2.8. Cell Lines and Cell Culture

We purchased cell lines, including 786-O, Caki-1 HK-2, A-498, and ACHN, from the American Type Culture Collection (ATCC, Manassas, VA, USA). For cell maintenance, we used Dulbecco’s modified Eagle’s growth medium (HyClone; GE Healthcare, Logan, UT, USA), which comprised fetal bovine serum (10%) (FBS; Gibco; Thermo Fisher Scientific, Inc., Waltham, MA, USA) and a solution of 1% penicillin–streptomycin as supplements. Incubation of the cells was set at 37 °C and 5% CO_2_.

### 2.9. Tissue Samples

A total of 33 paired human ccRCC and adjacent normal tissues were provided by the Union Hospital, Tongji Medical College (Wuhan, China), Department of Urology between 2018 and 2019. We stored the tissues in liquid nitrogen at −80 °C for subsequent extraction of RNA.

### 2.10. Immunohistochemistry

For immunohistochemical staining, we used four-micron sections obtained from C4-2 tissue embedded in paraffin. After sequential deparaffinization and rehydration, the tissues were incubated to retrieve the antigen upon 15-min incubation of the sections with H_2_O_2_ (3%) at room temperature. For blocking, we used the fetal bovine serum. Thereafter, we incubated these sections with the primary antibody at 4 °C, overnight. The immunoassay was carried out with a secondary antibody conjugated with HRP for 1 h. Finally, hematoxylin was used to stain the nucleus.

### 2.11. Western Blotting Assays

For this experiment, we lysed cells and tissues in RIPA protein lysis buffer (Beyotime Institute of Biotechnology, Haimen, China). The buffer comprised PMSF and protease inhibitor cocktail. Then, using the BCA kit (Beyotime Institute of Biotechnology), measurements of protein concentrations were taken following protocol stipulated by the manufacturer. Exactly 40 μg of protein was run on SDS-PAGE, subjected to gel electrophoresis for separation, then moved to polyvinylidene fluoride (PVDF) membranes. Using 5% skim milk (non-fat dried), the PVDF membranes were blocked at room temperature for 1 h. This was followed by incubating the membranes with primary antibodies at 4 °C overnight. Then, we performed a subsequent wash of the membranes followed by a 2-h incubation with secondary antibodies in blocking buffer at room temperature. Eventually, we sent samples to ChemiDoc-XRs+ (Bio-Rad Laboratories, Inc., Hercules, CA, USA) for protein visualization.

We used the antibodies GAPDH (Abclonal, A19056), ENO2 (Abclonal, A12341), HIF-1α (CST, #36169S) for Western blot analysis.

### 2.12. Extraction of RNA and Real-Time PCR Analysis

Following the manufacturer’s protocol, we used the TRizol reagent (Thermo, MA, USA) for RNA extraction from cells and tissues. RNA solution concentration and purity were determined using a NanoDrop 2000 spectrophotometer (NanoDrop Technologies, Wilmington, DC, USA). Using 1 μg of cell or tissue RNA, we reverse-transcribed the RNA. For qPCR analysis, the SYBR Green mix (YEASEN, Shanghai, China) was used in StepOnePlus™ Real-Time PCR System (Thermo Fisher Scientific, USA). Sample normalization was performed using *GAPDH*. Specific gene primer sequences are highlighted below.

*GAPDH* Forward 5′-GAGTCAACGGATTTGGTCGT-3′

Reverse 5′-GACAAGCTTCCCGTTCTCAG-3′

*ENO2* Forward 5′-GGGCACTCTACCAGGACTTT-3′

Reverse 5′-CAGACAGTTGCAGGCCTTTT-3′

*HIF1α* Forward 5′-TCCAAGAAGCCCTAACGTGT-3′

Reverse 5′-tgatcgtctggctgctgtaa-3′

### 2.13. Cell Viability Assay

Here, 2 × 10^3^ cell concentrations were transferred to 96-well plates for cell viability assay. CCK8 (YEASEN, Shanghai, China) was employed to detect the proliferation of cells at a concentration of 10% at 0, 24, 48, 72, and 96 h.

### 2.14. Transwell Assays

Before the experiment, a 24-h incubation of the cells with the serum-free medium was conducted. Then, after seeding the cells in the top chamber of the insert, they invaded through the Matrigel. At this point, we analyzed cell migration and invasion capabilities. The stained cells were photographed by a microscope, and the microscope randomly selected 10 fields for counting.

### 2.15. Chromatin Immunoprecipitation (ChIP)

ChIP analysis was performed utilizing the SimpleChIP^®^ Kit (Agarose Beads) (CST, 22188S, Boston, MA, USA) in accordance with the manufacturer’s instructions. Immunoprecipitation involved employing either rabbit anti-HIF1α antibody (#36169S, CST, Boston, USA) or normal rabbit IgG (#2729, CST, Boston, USA). The DNA obtained from the procedure underwent qPCR amplification to evaluate the interaction of *HIF1α* with the *ENO2* promoter region. The degree of enrichment was relatively normalized with respect to the IgG control.

### 2.16. Luciferase Reporter Assay

*ENO2* promoter regions were synthesized and cloned into the XhoI site of the GV354 vector (Genechem, Shanghai, China). A498 cells were seeded in 24-well plates and allowed to reach 70% confluence on the following day. The reporter plasmid (100 ng) was co-transfected with *ENO2* siRNA using Lipofectamine™ 3000. The luciferase activities of firefly and Renilla were measured using the dual luciferase reporter system (Promega, Madison, WI, USA).

### 2.17. In Vivo Experiment

Male athymic nude mice (aged 5 weeks) (Vital River, Beijing, China) were housed under standard conditions in a pathogen-free environment. A total of 12 mice were subcutaneously injected with A498 cells, either stable knockdown *ENO2* or control cells. Tumor growth was monitored regularly by measuring the dimensions using calipers, and tumor volume was calculated using the formula V = 0.5 × length × width^2^. All procedures involving animals were performed in compliance with ethical guidelines and approved by the Hubei Provincial Experimental Animal Research Centre.

### 2.18. Glucose, Lactate, and Adenosine Triphosphate (ATP) Detection

We used a glucose assay kit, lactate assay kit, and ATP assay kit (Jiancheng Company, Nanjing, China) to identify the utilization of glucose, production of lactate, and production of intracellular ATP, respectively, according to the manufacturer’s instructions.

### 2.19. Statistical Analysis

All statistical data analyses were conducted using SPSS Statistics 22.0 (IBM SPSS, Chicago, IL, USA) and Excel 2016 (Microsoft). Using the *t*-test, we analyzed the data of two groups. ANOVA analysis was used for the data with more than two groups. We generated the receiver operator characteristic (ROC) curves and area under the curve (AUC) to reveal the optimal cut off point yielding the highest total accuracy with respect to clarifying different clinical categorizations. With the Pearson correlation coefficient, we evaluated the correlation between two factors. Univariate and multivariate Cox proportional hazard regression analyses were used to determine the independent factor of ccRCC. We considered *p* < 0.05 as significant.

## 3. Results

### 3.1. Gene Expression Analysis Reveals the Significant Role of Glycolysis in ccRCC

Here, 626, 1529, and 1143 genes were extracted from the GSE36895, GSE66272, and GSE71963 datasets, respectively, using GEO2R. In order to confirm the reliability of DEGs in ccRCC, the three datasets were overlapped to obtain 425 DEGs (105 upregulated DEGs and 420 downregulated DEGs) (Appendix A). In order to explore the biological role of the DEGs, GO annotations and KEGG pathway enrichment analysis were performed using DAVID and clusterprofiler, respectively. GO annotations demonstrated that the upregulated DEGs were primarily concentrated in the cellular component (CC) and biological process (BP), including the extracellular space, extracellular region, response to hypoxia, and angiogenesis, while downregulated DEGs were also significantly enriched in cellular component (CC) and biological process (BP), including extracellular exosome, apical plasma membrane, excretion, and sodium ion homeostasis (Table 1). The association with metabolic processes, particularly involving fatty acid metabolism and glucose metabolism, is of significant relevance (Figure 1A). Through KEGG pathway analysis, we revealed that upregulated DEGs were mainly enriched in the HIF-1 signaling pathway, focal adhesion (Figure 1B), whereas downregulated DEGs were primarily enriched in drug metabolism, in collecting duct acid secretion (Appendix A). Among them, we found that glycolysis/gluconeogenesis and PPAR signaling pathways were enriched in the upregulation and downregulation of DEGs. This result suggests that glycolysis and the PPAR signaling pathway may exert vital roles in the occurrence and progression of ccRCC.

### 3.2. Establishing the PPI Network and Screening of Hub Genes

DEGs were analyzed for functional interaction through STRING. The DEGs’ PPI network was established by importing STRING analysis results into Cytoscape. The top three modules in the DEGs PPI network were extracted using the MCODE app in Cytoscape (Figure 2A–C). Genes in the modules were introduced into DAVID for GO annotation. The results showed that genes in module 1 with the highest PPI scores were mainly enriched in the biological processes (BP), including canonical glycolysis, glucose catabolic process to pyruvate, and the glucose metabolic process (Figure 2A). Notably, genes in module 2 were enriched in the biological processes (BP), including cell cycle, cell division, mitotic cell cycle (Figure 2B). Notably, in module 3, genes were enriched in the biological processes (BP), such as monovalent inorganic cation transport, sodium ion transport, and ion transport (Figure 2C). In addition, the top 30 genes with the highest degree of connectivity in the PPI network were chosen as hub genes using Cytoscape (Table 2). We introduced the hub genes into STRING to detect interactions between the encoded proteins. The result of GO annotation on the hub genes showed that it was mainly involved in the biological processes (BP), including platelet degranulation, blood vessel morphogenesis, and the establishment of localization (Figure 2D).

Through analyzing the PPI network of DEGs, we found that angiogenesis and glycolysis are significant factors affecting the progression of ccRCC. The important role of angiogenesis in ccRCC has been elucidated [29]. However, the influence of glycolysis on the occurrence and progression of ccRCC remains unclear. Therefore, glycolysis is considered a focus for future studies.

### 3.3. ENO2 Exhibits Significantly Elevated Expression in ccRCC

We conducted an overlap analysis of the DEGs with three independent datasets of glycolytic core genes (Beroukhim, GUMZ, Yusenko), leading to the identification of *ENO2* (Figure 3A). Then, through further exploration of the TCGA-KIRC database, we identified that *ENO2* was remarkably overexpressed in ccRCC (Figure 3B). The high *ENO2* expression in ccRCC was validated via five independent datasets from the Oncomine database (Figure 3C). Additionally, analysis of the ccRCC tissues were performed via immunohistochemical and qPCR techniques to determine the ENO2 protein and mRNA levels, respectively. Based on the findings, the ENO2 protein and mRNA levels detected in ccRCC tissues were significantly higher than in adjacent non-malignant tissues (Figure 3D,E). In comparison to the normal kidney cell line HK2, ENO2 expression was higher at the protein and mRNA levels in ccRCC cell lines (Figure 3F). Through Kaplan–Meier survival analysis, we elucidated the association of *ENO2* with ccRCC overall survival. The results showed that high *ENO2* expression has a correlation with worse overall survival of ccRCC (Figure 3G). Moreover, an analysis of Kaplan–Meier curves stratified by subgroups to determine overall survival and disease-free survival revealed that low expression of *ENO2* exhibited a high correlation with a poor prognosis (Appendix A). ROC curve analysis result revealed that *ENO2* exhibited good diagnostic value (Figure 3H). The clinical data of the TCGA-KIRC database were categorized into high and low expression groups based on the expression of *ENO2*. Two sets of data were imported into SPSS Statistics 22.0 according to different clinical–pathological parameters for univariate and multivariate Cox proportional hazard regression analysis. Of note, *ENO2* could be utilized as a potential independent ccRCC prognostic marker (Table 3 and Appendix A).

### 3.4. ENO2 Expression Correlates with Clinicopathological Parameters of Clear Cell Renal Cell Carcinoma

Clinicopathological parameters of ccRCC, including gender, age, T, N, M classification, TNM staging, and disease-free status were analyzed by chi-square test based on the level of *ENO2* expression to determine whether there was a correlation between the two. The results showed that *ENO2* expression was considerably associated with the clinical and pathological parameters of ccRCC (Appendix A). The TCGA-KIRC database was analyzed to reveal the specific association of *ENO2* expression with clinical parameters of ccRCC. We found that *ENO2* expression gradually increased with the advanced stage and grade of tumor (Appendix A). In summary, the higher expression of *ENO2* may indicate a higher tumor grade and a worse prognosis in ccRCC patients.

### 3.5. ENO2 Is Involved in Critical Biological Processes of Clear Cell Renal Cell Carcinoma and Correlates with Glycolysis

Epidermal growth factor receptor (*EGFR*), epidermal growth factor (*EGF*), vascular endothelial growth factor A (*VEGFA*), von Hippel–Lindau tumor suppressor (*VHL*), vascular endothelial growth factor B (*VEGFB*), and hypoxia-inducible factor 1 alpha (*HIF-1α*) are genes that have been proved by a large number of studies to exert crucial functions in ccRCC. The relationship between *ENO2* and ccRCC related genes was determined by analyzing how each gene is expressed in the TCGA-KIRC database. Related gene expression heatmap results demonstrated that the expression of *ENO2* has a positive correlation with *EGFR*, *VEGFA*, and *VEGFB*, but is negatively correlated with *EGF*, *VHL*, and *HIF-1α* (Figure 4A, Appendix A). In order to further clarify the way in which *ENO2* plays a role in ccRCC, the first neighbors of *ENO2* in the DEGs’ PPI network were screened out through Cytoscape software (3.7.2). The screened genes were explored via GO annotation and KEGG pathway analysis using DAVID (Figure 4B) and clusterprofiler (Appendix A), respectively. GO annotation results showed that *ENO2* and its first neighbors were mainly enriched in biological processes, including glucose metabolic process, canonical glycolysis, and glucose catabolic process to pyruvate. KEGG pathway analysis results indicated that it was mainly associated with glycolysis/gluconeogenesis and the HIF-1 signaling pathway. Based on the aforementioned findings, we have reasonable grounds to infer that *ENO2* might be regulated by *HIF-1α* in ccRCC. To investigate this, we generated a HIF1A knockdown ccRCC cell line through siRNA transfection (Appendix A). Western blot and qRT-PCR analyses indicated that HIF-1α knockdown led to a decrease in ENO2 protein and mRNA levels (Figure 4C,D). Acting as a transcription factor, HIF-1α activates target gene transcription by binding to hypoxia-response elements (HREs) within target gene promoter regions [30]. Within the proximal 1000 bp region of the *ENO2* promoter, we identified three putative *HIF-1α* binding sites (Figure 4E). ChIP experiments conducted in A498 cells confirmed *HIF-1α* binding to site1, site2, and site3 (Figure 4F). Utilizing truncated plasmids constructed based on these three binding sites, transfection experiments in *HIF-1α* knockdown A498 cells, coupled with dual luciferase reporter gene assays, revealed distinct outcomes for site3 compared to site1 and site2. Notably, site3 exhibited a capacity to reverse the reduced fluorescence intensity caused by *HIF-1α* knockdown (Figure 4G), implying that *HIF-1α* binding to site3 within the *ENO2* promoter region activates *ENO2* transcription.

### 3.6. ENO2 Promotes Glucose Metabolism and ccRCC Progression

With these findings, it was suggested that the expression of *ENO2* might influence ccRCC progression. To validate this assumption, *ENO2* was knocked down by siRNA in A498 and 786-O cell lines (Figure 5A,B). Glucose utilization, lactate detection, and intracellular ATP assays demonstrated that knockdown of *ENO2* inhibited glucose utilization (Figure 5C), lactate production (Figure 5D), and intracellular ATP generation (Figure 5E) in A498 and 786-O cells. The CCK8 assays were employed to assess ccRCC cell proliferation after knocking down *ENO2*. Results showed that *ENO2* knockdown in the ccRCC cell lines remarkably lowered the rate of proliferation of ccRCC cells (Figure 5F,G). Then, invasion and migration, as key indicators of tumor progression, were further carried out. Transwell was conducted to reevaluate how *ENO2* expression affects invasion and migration ability in ccRCC cells. We found that the *ENO2* downregulation blocked off ccRCC cell migration and invasion (Figure 5H,I). We established a subcutaneous tumor model in nude mice by separately injecting A498 cells infected with sh*ENO2* or control lentivirus (Figure 5J). This was aimed at further validating the impact of *ENO2* on ccRCC growth in vivo. Depletion of *ENO2* significantly suppressed both the weight (Figure 5K) and volume (Figure 5L) of subcutaneous tumors in nude mice, underscoring *ENO2*’s role in promoting ccRCC growth within the in vivo context. Through functional experiments, we found that *ENO2* regulates glucose metabolism and exerts oncogenic functions in ccRCC.

### 3.7. CcRCC Patients with High ENO2 Expression Have Higher Levels of Tumor Mutation Burden (TMB)

To investigate the association between *ENO2* expression and internal gene mutations in ccRCC, we obtained whole-exome sequencing data of ccRCC patients from the TCGA database and used Maftools to summarize the mutation data. Patients were divided into high and low *ENO2* expression subgroups based on their *ENO2* expression levels. By evaluating the levels of TMB in the two groups of patients, we found that those with high *ENO2* expression had higher TMB (Figure 6A). Further classification of the mutations was conducted using a variant effect predictor, and the frequency of consistent missense mutations was highest in both groups (Figure 6B and Appendix A). Among all types of mutations, the frequency of SNP occurrence was the highest (Figure 6C and Appendix A). Similarly, the most frequent SNV type in both groups of mutations was C > T transition (Figure 6D and Appendix A). We further investigated the gene mutations in the two patient groups and described the common mutated genes in patients with high and low *ENO2* expression (Figure 6E,F). Among all these genes, the mutation rates of *VHL*, *PBRM1*, *TTN*, *SETD2*, and *BAP1* were both at a peak in the two groups of patients (>10%) (Figure 6G). We then examined the enrichment of common oncogenic signaling pathways in the high and low *ENO2* expression subgroups. By comparison, we found that the proportion of samples affected by the PI3K pathway was higher in the high *ENO2* expression subgroup than in the low-risk subgroup (Figure 6H and Appendix A), suggesting that the PI3K pathway is particularly prominent in the high-risk subgroup. We conducted GSEA enrichment analysis to compare the high and low TMB groups and found that the enrichment results were mainly associated with glucose metabolism (Figure 6I).

## 4. Discussion

Here, we have elucidated *ENO2* as a pivotal gene regulating the metabolic reprogramming of ccRCC through a series of bioinformatics analyses. *ENO2* is upregulated in ccRCC and serves as a potential biomarker for this condition. Functional and mechanistic investigations have corroborated that *ENO2* is transcriptionally activated by *HIF-1α*, and it orchestrates tumor cell glycolysis to sustain the energy supply and promote the progression of ccRCC.

Previous investigations have identified numerous biomarkers for ccRCC. However, the clinical significance of most remains to be established. Through differential expression analyses of multiple ccRCC gene expression datasets, we have revealed that in addition to its angiogenic role, glycolysis constitutes another pivotal factor influencing ccRCC initiation and progression. The Warburg effect stands as one of the earliest manifestations of metabolic reprogramming in ccRCC [31]. Despite ample oxygen availability, a majority of cancer cells predominantly generate energy through glycolysis, in stark contrast to most normal cells that primarily utilize mitochondrial oxidative phosphorylation. This tumor-specific Warburg effect furnishes the energy and biosynthetic precursors requisite for propelling tumorigenesis [32,33]. The Warburg effect exerts its influence across various facets of ccRCC tumorigenesis, encompassing tumor growth, survival, angiogenesis, and metastasis [34,35,36]. Moreover, the acidic microenvironment resultant from lactate accumulation within the tumor milieu fosters invasive behavior and immune evasion [37,38].

ENO2 plays a pivotal role in energy metabolism and contributes to the metabolic reprogramming and progression of tumors [39]. Enhanced glycolytic activity empowers cancer cells to rapidly generate energy and biosynthetic intermediates, thereby supporting their growth and survival. ENO2, by catalyzing a crucial step in glycolysis, promotes the conversion of 2-phosphoglycerate to phosphoenolpyruvate, thereby aiding in this metabolic adaptation [40]. *ENO2* has been upregulated in various tumor types, including head and neck cancer [39], colorectal cancer [41], pancreatic cancer [42], and bladder cancer [43], and has been associated with more aggressive phenotypes. Our investigations further confirm that heightened *ENO2* expression correlates with elevated glucose uptake and lactate production, hallmark features of the Warburg effect. This observation underscores the role of *ENO2* in sustaining the growth of tumors, including ccRCC. These findings shed light on the significance of ENO2 as a key regulator of tumor metabolism.

In the landscape of ccRCC, the HIF pathway assumes paramount importance. Activated in response to VHL gene mutations frequently observed in ccRCC, HIF engages with the 5′-(A/G)CGTG-3′ hypoxia-response elements (HREs) within target gene promoters, instigating their transcriptional activation [44,45,46]. Hypoxia stands as a predominant component of solid tumors, primarily arising from compromised angiogenesis post-rapid tumor expansion, thereby yielding pathophysiological microcirculatory perturbations. *HIF-1α* assumes a critical role in orchestrating tumor metabolism to respond to the increasingly hypoxic microenvironment [47]. Oxygen limitation, a known driver of glycolysis, necessitates reliance on glycolysis for ATP production due to limited oxidative phosphorylation. *HIF-1α* is pivotal in this process, inducing the expression of glycolytic enzymes, such as hexokinase and phosphofructokinase, thereby sustaining ATP generation [48,49]. Notably, existing research has indicated a positive regulatory relationship between *HIF-1α* and *ENO2*. Fang et al. demonstrated that hypoxia upregulates the expression of both *HIF-1α* and *ENO2* in human myeloid cells [50]. In a related context, Leiherer et al. confirmed the interaction between *HIF-1α* and HREs of ENO2, a key enzyme in glycolytic metabolism, within adipocytes [30]. These studies highlight the intricate interplay between *HIF-1α* and *ENO2* across diverse cellular contexts. Our investigation further underscores the *HIF-1α*-mediated transcriptional regulation of *ENO2* in ccRCC, emphasizing the pivotal role of *HIF-1α* in metabolic control.

In conclusion, the research focused on the role of *ENO2* in ccRCC has illuminated its pivotal involvement in the intricate landscape of tumor metabolism and progression. Through its participation in glycolysis, *ENO2* emerges as a key player in fueling the energy demands of ccRCC cells, ultimately contributing to their rapid proliferation and survival. Furthermore, the identification of *HIF-1α*-mediated transcriptional regulation of *ENO2* adds a crucial layer to our understanding of the metabolic adaptations crucial for tumor growth in the context of ccRCC. The connection between *ENO2* and the HIF pathway underscores the sophisticated interplay between metabolic reprogramming and the tumor microenvironment, highlighting the significance of *ENO2* in translating the hypoxic response into altered glycolytic activity. As such, targeting *ENO2* and its interactions with *HIF-1α* could potentially offer novel therapeutic avenues for ccRCC treatment by exploiting the vulnerabilities that arise from metabolic adaptations. Overall, these findings underscore the importance of unraveling the molecular intricacies governing ccRCC metabolism, paving the way for potential advancements in both diagnosis and therapeutic strategies targeting this challenging malignancy.

## Figures and Tables

**Figure 1 biomedicines-11-02499-f001:**
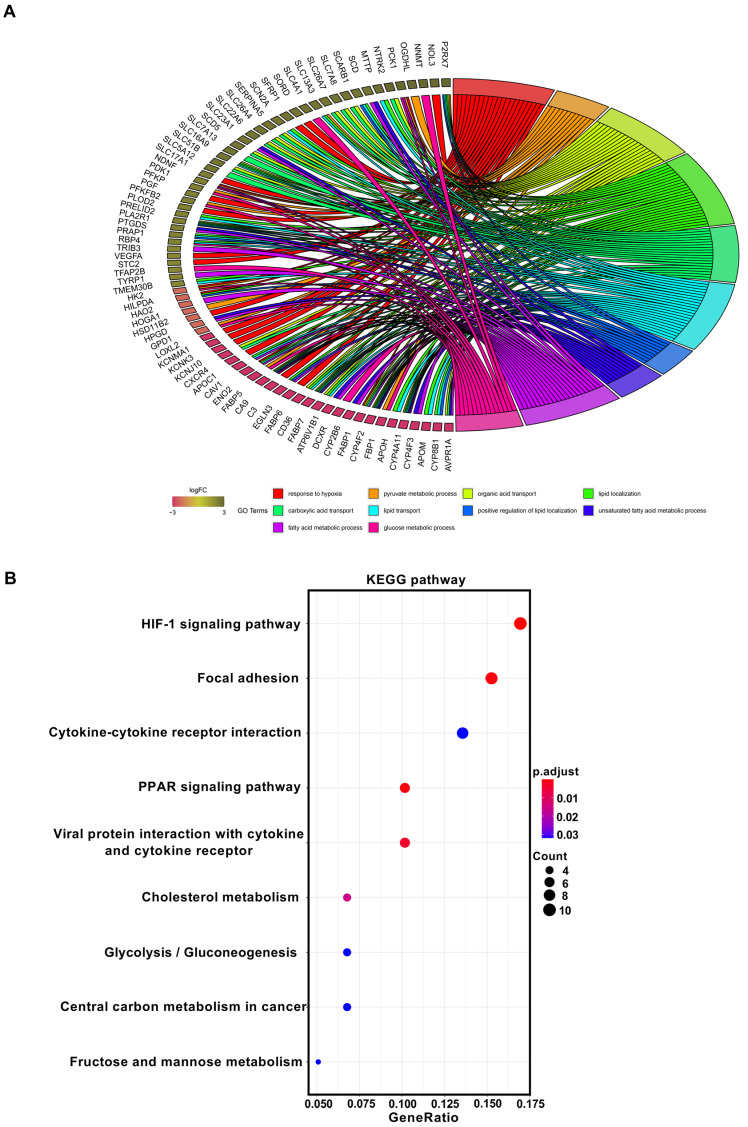
KEGG and GO pathway analyses. (**A**) The top 10 enriched GO terms. GO: Gene Ontology. (**B**) Analysis of the upregulated KEGG pathway.

**Figure 2 biomedicines-11-02499-f002:**
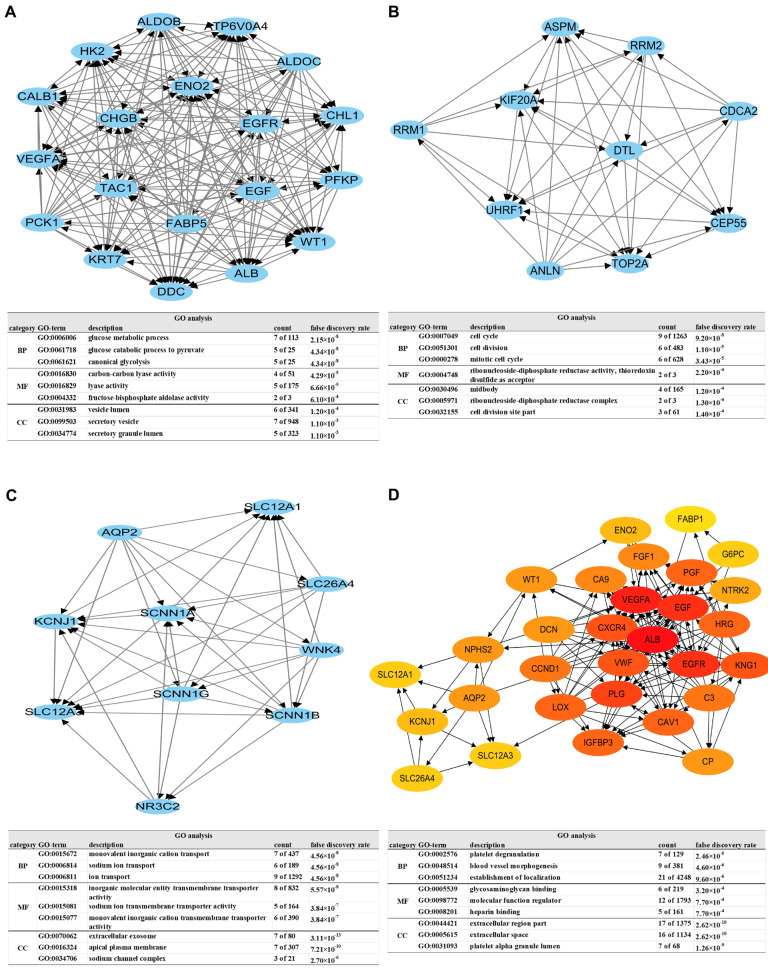
PPI network analysis and screening for hub genes. Using the MCODE tool of Cytoscape software, the top 3 modules in the PPI network were identified. (**A**) Module 1 and its GO annotations; (**B**) module 2 and its GO annotations; (**C**) module 3 and its GO annotations. (**D**) The 30 selected hub genes were constructed into a new PPI network using STRING and Cytoscape.

**Figure 3 biomedicines-11-02499-f003:**
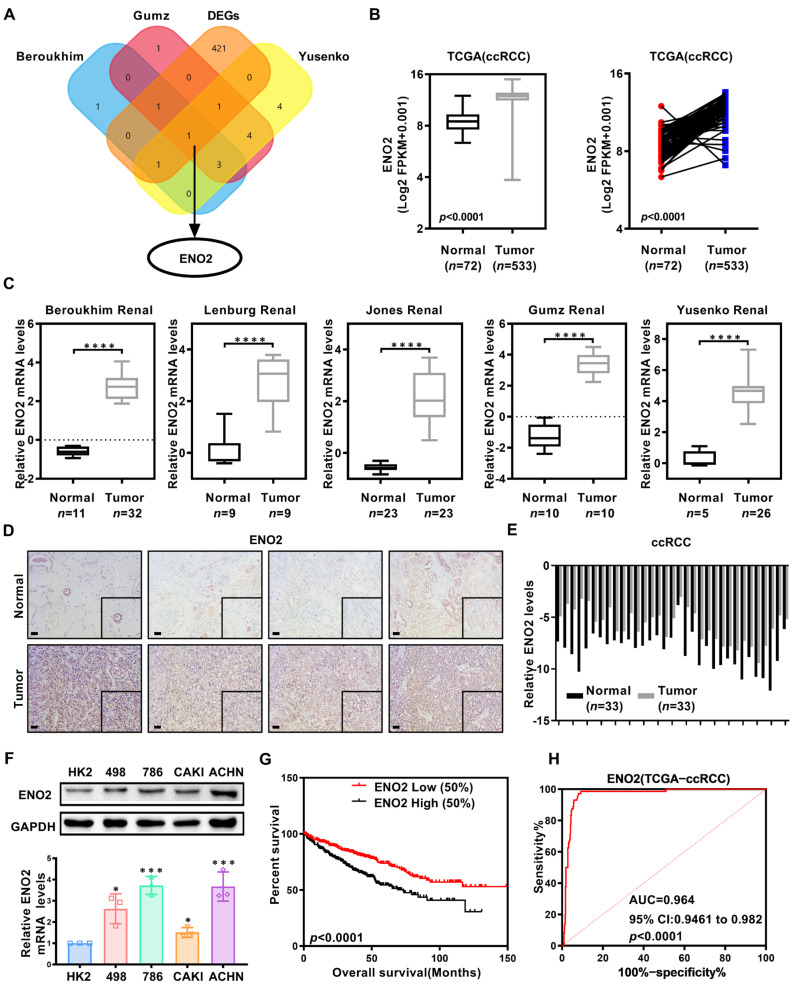
Data mining in Oncomine database and TCGA kidney clear cell carcinoma (TCGA-KIRC). (**A**) A Venn diagram showing 3 independent glycolysis metabolism-related gene sets retrieved from the Oncomine database (https://www.oncomine.org (accessed on 10 March 2020)) and DEGs. (**B**) Levels of the *ENO2* mRNAs in ccRCC tissues and paired tissues from patients with ccRCC based on data from the TCGA database. (**C**) The expression of *ENO2* in five independent additional gene sets from the Oncomine database; *t*-test, **** indicates *p* < 0.0001. (**D**) Immunohistochemical (IHC) staining was performed for *ENO2* in 4 pairs of ccRCC and adjacent non-malignant tissues (scale bar: 50 μm). (**E**) A total of 33 pairs of ccRCC patients’ tumor tissues and adjacent non-malignant tissues of *ENO2* mRNA levels; *t* test, *p* <0.0001. (**F**) ENO2 protein and mRNA expression difference between a normal kidney cell line and four ccRCC cell lines; *t*-test, * indicates *p* < 0.05, *** indicates *p* < 0.001. (**G**) The Kaplan–Meier curves of *ENO2* expression in patients with ccRCC for determining overall survival (OS). The *p* value is obtained through Log-rank (Mantel–Cox) test. (**H**) The ROC curves of *ENO2* (AUC = 0.964 95% CI: 0.9461 to 0.982; *p* < 0.0001).

**Figure 4 biomedicines-11-02499-f004:**
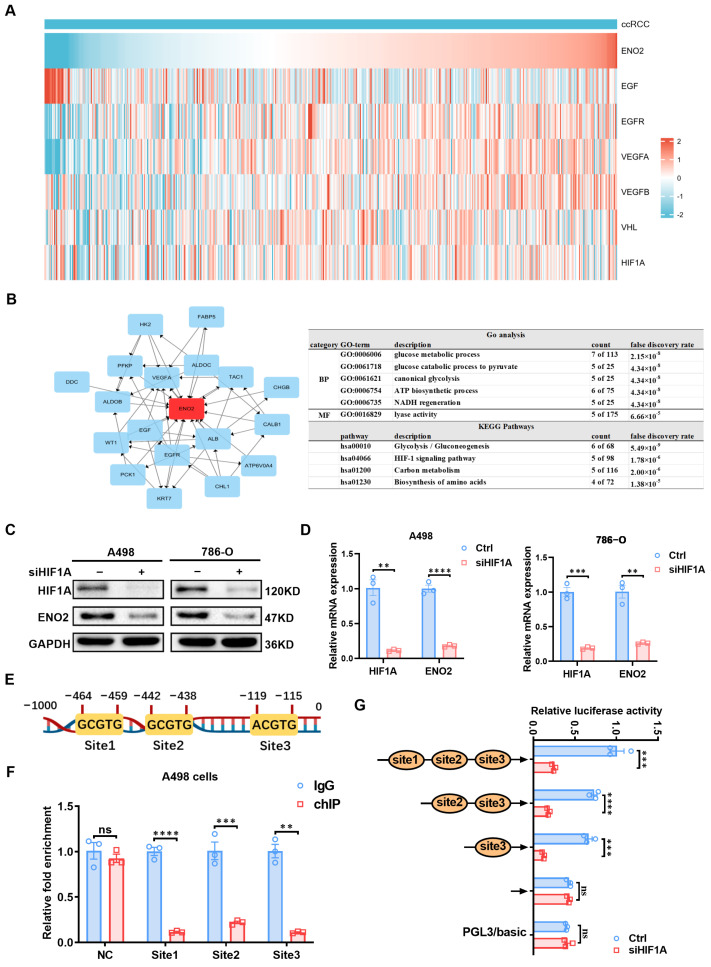
*ENO2* is involved in key biological processes in ccRCC and is correlated with glycolysis. The results represent the mean ± SEM of three independent experiments with at least three replicates per experiment. **** *p* < 0.0001, *** *p* < 0.001, and ** *p* < 0.01. (**A**) The correlation heatmap between *ENO2* and *EGF*, *EGFR*, *VEGFA*, *VEGFB*, *VHL*, and *HIF-1α* expression in the TCGA-KIRC database. (**B**) PPI network of the first neighbor of ENO2 in the DEGs’ PPI network and its GO annotation and KEGG pathway analysis results. (**C**) ENO2 protein levels after HIF-1α knockdown in ccRCC cells. (**D**) mRNA levels of *ENO2* after *HIF-1α* knockdown in ccRCC cells. (**E**) HRE binding sites in the *ENO2* promoter region. (**F**) ChIP results based on HRE binding sites in the *ENO2* promoter region. (**G**) Fluorescence intensity of dual luciferase reporter genes in truncated plasmids.

**Figure 5 biomedicines-11-02499-f005:**
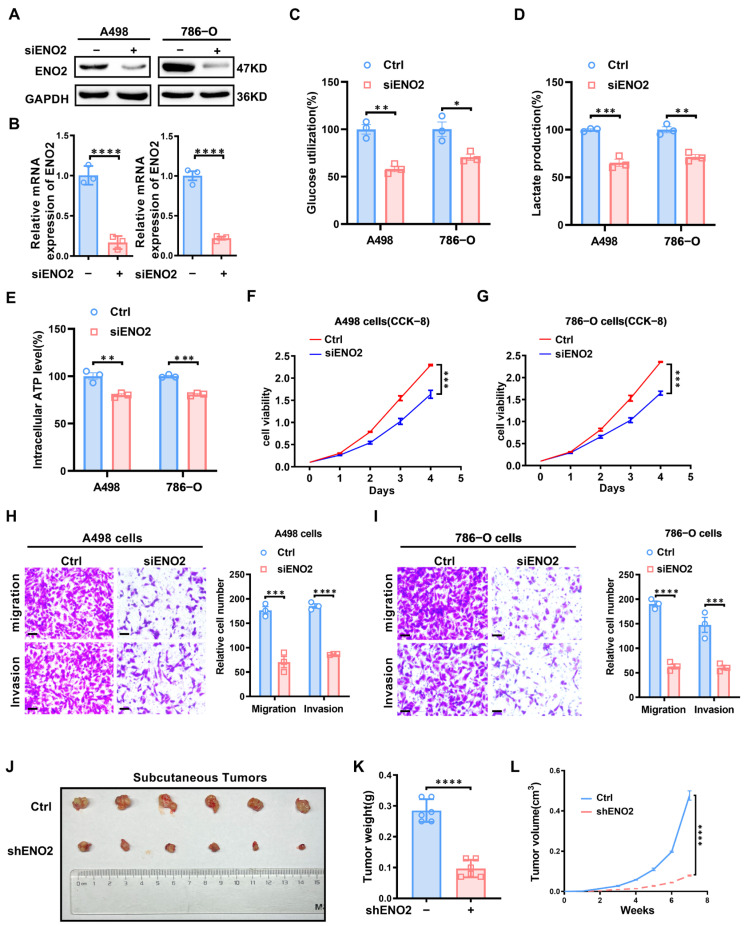
Knockdown of *ENO2* in ccRCC cells can impede ccRCC progression. CcRCC cell lines were transfected with siRNA to knock down *ENO2*. Results are the mean ± SE of three independent experiments. **** indicates *p* < 0.0001, *** indicates *p* < 0.001, ** indicates *p* < 0.01, and * indicates *p* < 0.05. (**A**) Western blot assay verifying the ENO2 knockdown in A498 and 786-O cell lines at the protein level. (**B**) qPCR was used to verify *ENO2* knockdown in A498 and 786-O cell lines at the mRNA level. (**C**–**E**) The impact of *ENO2* knockdown on glucose utilization, lactate production, and intracellular ATP generation was evaluated in A498 and 786-O cells using glucose assay, lactate assay, and intracellular ATP assay, respectively. (**F**,**G**) CCK8 assays used in determining the cell growth of A498 and 786-O cell lines after *ENO2* knockdown. (**H**) Transwell assay was employed to evaluate the migratory and invasive capabilities of A498 cells following *ENO2* knockdown (scale bar: 50 μm). (**I**) Transwell assay was employed to evaluate the migratory and invasive capabilities of 786-O cells following *ENO2* knockdown (scale bar: 50 μm). (**J**) Subcutaneous tumor images obtained by euthanizing nude mice 7 weeks after subcutaneous injection of A498 cells infected with shENO2 or control lentivirus. (**K**) Subcutaneous tumor weight statistics in nude mice. (**L**) Growth curve of subcutaneous tumor volume in nude mice.

**Figure 6 biomedicines-11-02499-f006:**
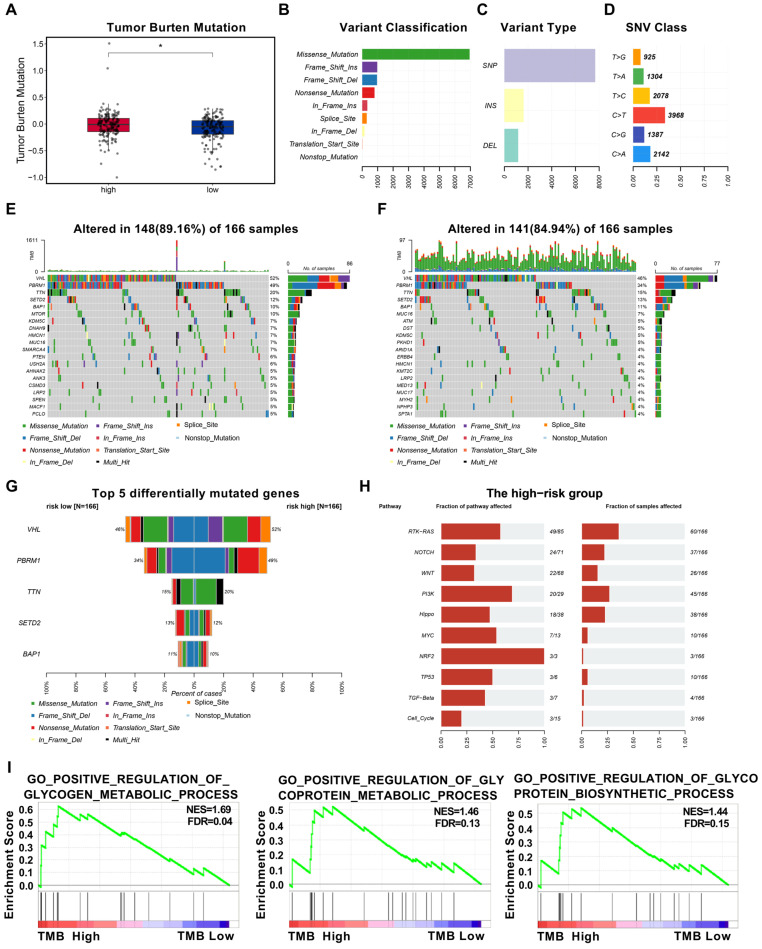
CcRCC patients with high *ENO2* expression have higher levels of tumor mutation burden (TMB). * indicates *p* < 0.05. (**A**) Comparison of TMB between high-expression and low-expression subgroups based on *ENO2*. (**B**,**C**) Gene mutation categories in the high-expression subgroup of *ENO2*. (**D**) Classification of SNVs in the high-expression subgroup of *ENO2*. (**E**,**F**) Waterfall plot of common somatic mutations in tumor cells in the high-expression and low-expression subgroups of *ENO2*. (**G**) Top 5 differentially mutated genes between the high-expression and low-expression subgroups of *ENO2*. (**H**) Top 10 signaling pathways enriched with mutated genes in the high-expression subgroup of *ENO2*. (**I**) Results of GSEA enrichment.

**Table 1 biomedicines-11-02499-t001:** The top five enriched GO terms of the DEGs.

	Category	ID	Term	Count	*p*-Value
Upregulated DEGs (*n* = 105)	BP	GO:0001666	Response to hypoxia	12	4.39 × 10^−9^
		GO:0001525	Angiogenesis	10	5.68 × 10^−6^
		GO:0007595	Lactation	5	9.90 × 10^−5^
		GO:0051592	Response to calcium ion	5	3.50 × 10^−4^
		GO:0061621	Canonical glycolysis	4	4.41 × 10^−4^
	CC	GO:0005615	Extracellular space	27	1.60 × 10^−8^
		GO:0005576	Extracellular region	26	1.99 × 10^−6^
		GO:0005581	Collagen trimer	6	1.66 × 10^−4^
		GO:0005901	Caveola	5	4.82 × 10^−4^
		GO:0005578	Proteinaceous extracellular matrix	7	0.003969211
	MF	GO:0042803	Protein homodimerization activity	13	8.73 × 10^−4^
		GO:0005102	Receptor binding	8	0.003999188
		GO:0046982	Protein heterodimerization activity	9	0.00499099
		GO:0004720	Protein-lysine 6-oxidase activity	2	0.016964763
		GO:0005178	Integrin binding	4	0.022013844
Downregulated DEGs (*n* = 320)	CC	GO:0070062	Extracellular exosome	127	8.66 × 10^−29^
		GO:0016324	Apical plasma membrane	32	1.31 × 10^−16^
		GO:0016323	Basolateral plasma membrane	20	1.56 × 10^−10^
		GO:0005887	Integral component of plasma membrane	58	2.38 × 10^−10^
		GO:0005886	Plasma membrane	106	7.00 × 10^−7^
	BP	GO:0007588	Excretion	11	5.16 × 10^−10^
		GO:0055078	Sodium ion homeostasis	7	1.15 × 10^−8^
		GO:0035725	Sodium ion transmembrane transport	11	5.29 × 10^−7^
		GO:0006814	Sodium ion transport	11	1.41 × 10^−6^
		GO:0055114	Oxidation–reduction process	29	1.80 × 10^−6^
	MF	GO:0015301	Anion:anion antiporter activity	7	1.57 × 10^−6^
		GO:0016491	Oxidoreductase activity	14	3.04 × 10^−6^
		GO:0005215	Transporter activity	14	3.38 × 10^−5^
		GO:0042803	Protein homodimerization activity	28	8.69 × 10^−5^
		GO:0008201	Heparin binding	11	3.44 × 10^−4^

**Table 2 biomedicines-11-02499-t002:** Top 30 hub genes with degree > 10.

Hub Genes	Degree of Connectivity
ALB	82
VEGFA	58
EGFR	56
EGF	50
KNG1	32
C3	30
CXCR4	27
AQP2	27
PLG	26
CCND1	25
KCNJ1	24
SLC12A1	24
SLC12A3	23
CAV1	23
HRG	23
VWF	22
LOX	22
SLC26A4	21
IGFBP3	19
CP	19
DCN	19
PGF	18
FGF1	18
NTRK2	18
NPHS2	18
FABP1	18
ENO2	18
CA9	17
WT1	17
G6PC	17

**Table 3 biomedicines-11-02499-t003:** Univariate and multivariate analyses of *ENO2* mRNA level and patient overall survival (OS).

Univariate Analysis				Multivariate Analysis ^c^		
Variable	HR ^a^	95%CI ^b^	*p*	HR	95% CI	*p*
Overall survival						
Age (years)						
≤60 (*n* = 267)	1.817	1.336−2.471	0.000 *	1.942	1.254−3.007	0.003 *
>60 (*n* = 270)						
Gender						
Female (*n* = 191)	0.750	0.699−1.294	0.951			
Male (*n* = 346)						
T stage						
T1 or T2 (*n* = 344)	3.217	2.375−4.358	0.000 *	2.208	1.419−3.436	0.000 *
T3 or T4 (*n* = 193)						
N stage						
N0 (*n* = 239)	3.604	1.954−6.647	0.000 *			
N1 (*n* = 17)						
M stage						
M0 (*n* = 425)	4.400	3.224−6.006	0.000 *	3.430	2.149−5.473	0.000 *
M1 (*n* = 79)						
G grade						
G1 or G2 (*n* = 239)	2.741	1.946−3.859	0.000 *	1.752	1.068−2.876	0.026 *
G3 or G4 (*n* = 280)						
TNM stage						
I + II (*n* = 326)	3.931	2.862−5.399	0.000 *			
III + IV (*n* = 208)						
ENO2						
Low (*n* = 267)	1.837	1.352−2.495	0.000 *	1.891	1.235−2.895	0.003 *
High (*n* = 270)						

^a^ Hazard ratio, estimated from Cox proportional hazard regression model. ^b^ Confidence interval of the estimated HR. ^c^ Multivariate models were adjusted for T, M classification, and G stage and age. The asterisk (*) indicates statistical significance.

## Data Availability

The material supporting the conclusion of this review has been included within the article.

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
