# Peer review of "ENO2 as a Biomarker Regulating Energy Metabolism to Promote Tumor Progression in Clear Cell Renal Cell Carcinoma"

_biomedicines, 2023, doi:10.3390/biomedicines11092499_

Round 1

Reviewer 1 Report

The article is devoted to an actual topic. The use of modern approaches is noted. However, it is worth reworking the section discussion of the results obtained in order to find approaches for practical application.

The article is devoted to an actual topic. The use of modern approaches is noted. However, it is worth reworking the section discussion of the results obtained in order to find approaches for practical application.

Author Response

We greatly appreciate your time and effort in evaluating our manuscript and providing constructive suggestions to further enhance its quality. Below, we present our point-to-point responses.

The article is devoted to an actual topic. The use of modern approaches is noted. However, it is worth reworking the section discussion of the results obtained in order to find approaches for practical application.

We are deeply appreciative of the meticulous feedback provided by the reviewer on our manuscript, and we fully agree with the reviewer's comments. We have revised the Discussion section of the manuscript. In the first paragraph, we delineate the outcomes of our study, while the concluding paragraph delves into a comprehensive discussion of ENO2's role in the initiation and progression of ccRCC. Furthermore, we explore the potential for therapeutic interventions targeting ENO2 and its upstream regulator HIF1α in the treatment of ccRCC. The revised "Discussion" section is as follows:

Here, we have elucidated ENO2 as a pivotal gene regulating the metabolic reprogramming of ccRCC through a series of bioinformatics analyses. ENO2 is upregulated in ccRCC and serves as a potential biomarker for this condition. Functional and mechanistic investigations have corroborated that ENO2 is transcriptionally activated by HIF1α, and it orchestrates tumor cell glycolysis to sustain the energy supply and promote the progression of ccRCC.

Previous investigations have identified numerous biomarkers for ccRCC. However, the clinical significance of most remains to be established. Through differential expression analyses of multiple ccRCC gene expression datasets, we have revealed that in addition to its angiogenic role, glycolysis constitutes another pivotal factor influencing ccRCC initiation and progression. The Warburg effect stands as one of the earliest manifestations of metabolic reprogramming in ccRCC (31). Despite ample oxygen availability, a majority of cancer cells predominantly generate energy through glycolysis, in stark contrast to most normal cells that primarily utilize mitochondrial oxidative phosphorylation. This tumor-specific Warburg effect furnishes the energy and biosynthetic precursors requisite for propelling tumorigenesis (32,33). The Warburg effect exerts its influence across various facets of ccRCC tumorigenesis, encompassing tumor growth, survival, angiogenesis, and metastasis (34-36). Moreover, the acidic microenvironment resultant from lactate accumulation within the tumor milieu fosters invasive behavior and immune evasion (37,38).

ENO2 plays a pivotal role in energy metabolism and contributes to the metabolic reprogramming and progression of tumors (39). Enhanced glycolytic activity empowers cancer cells to rapidly generate energy and biosynthetic intermediates, thereby supporting their growth and survival. ENO2, by catalyzing a crucial step in glycolysis, promotes the conversion of 2-phosphoglycerate to phosphoenolpyruvate, thereby aiding in this metabolic adaptation (40). ENO2 has been upregulated in various tumor types, including head and neck cancer (39), colorectal cancer (41), pancreatic cancer (42), and bladder cancer (43), and has been associated with more aggressive phenotypes. Our investigations further confirm that heightened ENO2 expression correlates with elevated glucose uptake and lactate production, hallmark features of the Warburg effect. This observation underscores the role of ENO2 in sustaining the growth of tumors, including ccRCC. These findings shed light on the significance of ENO2 as a key regulator of tumor metabolism.

In the landscape of ccRCC, the HIF pathway assumes paramount importance. Activated in response to VHL gene mutations frequently observed in ccRCC, HIF engages with the 5′-(A/G)CGTG-3′ hypoxia response elements (HREs) within target gene promoters, instigating their transcriptional activation (44-46). Hypoxia stands as a predominant component of solid tumors, primarily arising from compromised angiogenesis post rapid tumor expansion, thereby yielding pathophysiological microcirculatory perturbations. HIF1α assumes a critical role in orchestrating tumor metabolism to respond to the increasingly hypoxic microenvironment (47). Oxygen limitation, a known driver of glycolysis, necessitates reliance on glycolysis for ATP production due to limited oxidative phosphorylation. HIF1α is pivotal in this process, inducing the expression of glycolytic enzymes such as hexokinase and phosphofructokinase, thereby sustaining ATP generation (48,49). Notably, existing research has indicated a positive regulatory relationship between HIF1α and ENO2. Fang et al. demonstrated that hypoxia upregulates the expression of both HIF1α and ENO2 in human myeloid cells (50). In a related context, Leiherer et al. confirmed the interaction between HIF1α and HREs of enolase 2 (ENO2), a key enzyme in glycolytic metabolism, within adipocytes (30). These studies highlight the intricate interplay between HIF1α and ENO2 across diverse cellular contexts. Our investigation further underscores the HIF1α-mediated transcriptional regulation of ENO2 in ccRCC, emphasizing the pivotal role of HIF1α in metabolic control.

In conclusion, the research focused on the role of ENO2 in ccRCC has illuminated its pivotal involvement in the intricate landscape of tumor metabolism and progression. Through its participation in glycolysis, ENO2 emerges as a key player in fueling the energy demands of ccRCC cells, ultimately contributing to their rapid proliferation and survival. Furthermore, the identification of HIF1α-mediated transcriptional regulation of ENO2 adds a crucial layer to our understanding of the metabolic adaptations crucial for tumor growth in the context of ccRCC. The connection between ENO2 and the HIF pathway underscores the sophisticated interplay between metabolic reprogramming and the tumor microenvironment, highlighting the significance of ENO2 in translating the hypoxic response into altered glycolytic activity. As such, targeting ENO2 and its interactions with HIF1α could potentially offer novel therapeutic avenues for ccRCC treatment by exploiting the vulnerabilities that arise from metabolic adaptations. Overall, these findings underscore the importance of unraveling the molecular intricacies governing ccRCC metabolism, paving the way for potential advancements in both diagnosis and therapeutic strategies targeting this challenging malignancy.

Reviewer 2 Report

In this study, the authors investigated the role of ENO2 in clear cell renal cell carcinoma (ccRCC). The data showed that ENO2 overexpressed in ccRCC that may contribute to glycolysis. Thus, ENO2 could be a potential biomarker for the diagnosis of ccRCC. Overall, this study is interesting. My main concern is the novelty. There is a similar study was published previously (Pan et al., Disease Markers, 2022, Integrated Analysis of the Role of Enolase 2 in Clear Cell Renal Cell Carcinoma). The authors should provide either more solid evidences or new mechanisms to strengthen your conclusions and novelty. My specific comments are listed below.

1.     The authors analyzed three GEO datasets and found HIF-1 signaling pathway, including glycolysis signaling pathway was mainly enriched in up-regulation of DEGs. However, why only one glycolysis-related gene ENO2 was enriched? In other words, glycolysis signaling pathway as a main enriched in up-regulation of DEGs, it should be more glycolysis-related genes in ccRCC.

2.     For all micrograghs, the scale bars are required (Fig.3D, Fig.5H). For immunoblots (Fig.3F, Fig.5A), the molecular markers should be shown.

3.     In Figure 5B, the bar graghs should be shown as other ones (Figure 5C, D, E) with individual data points.

4.     In ccRCC, HIFs are activated. The mechanism of HIFs regulating their target genes (Yang et al, Nat Commun, 2022 Jan 14;13(1):316; Zhang et al., Science, 2018 Jul 20;361(6399):290-295.) and the relationship of ENO2 and HIFs (Leiherer et al., Mol Cell Endocrinol. 2014 Mar 5; 383(1-2): 21–31; Fang et al., Blood 2009 Jul 23;114(4):844-59.) should be mentioned either in the “Introduction” or “Discussion” section. But, it’s better to confirm this by experiments in one of the ccRCC cell lines. Is ENO2 a direct HIFs target gene? Does HIFs bind to ENO2 promoter?

5.     It would be better to validate the function of ENO2 in clear cell renal cell carcinoma in vivo animal model.

6.     The writing of this manuscript need to be improved. For example, there are several typos, including but not limited to: (a) “…including HIF-2a and HIF-1a, are activated 0”; (b) Figure 5F, “time (days)”; (c) 5H, asterisks.

Please carefully revise the manuscript. Many aspects of the writing of this manuscript need to be improved. For example, there are several typos, including but not limited to: (a) “…including HIF-2a and HIF-1a, are activated 0”; (b) Figure 5F, “time (days)”; (c) 5H, asterisks.

Author Response

We are grateful for your time and effort in evaluating our manuscript and providing constructive suggestions to further enhance its quality. Below, we present our point-to-point responses.

  1. The authors analyzed three GEO datasets and found HIF-1 signaling pathway, including glycolysis signaling pathway was mainly enriched in up-regulation of DEGs. However, why only one glycolysis-related gene ENO2 was enriched? In other words, glycolysis signaling pathway as a main enriched in up-regulation of DEGs, it should be more glycolysis-related genes in ccRCC.

We greatly appreciate the valuable suggestions provided by the reviewers. In response to the reviewers' inquiries, we offer the following explanations:

(1) Tumors exhibit pronounced heterogeneity, wherein the same pathway can lead to differential gene expression across distinct individuals within the same tumor. Theoretically, a higher intersection of sequencing data from various individuals can mitigate this heterogeneity to a greater extent. To identify commonalities within the same tumor context, we specifically selected glycolysis-related datasets directly associated with clear cell renal cell carcinoma (ccRCC) from the Oncomine database. By leveraging data from multiple diverse sources, we aimed to mitigate such inter-individual differences. Consequently, the overall filtering outcomes are highly concentrated, thereby excluding numerous elements lacking commonality. Concurrently, in our selection of glycolysis-related datasets, we focused on core genes rather than affiliated genes, resulting in a noticeable reduction in the number of genes in the dataset. The specific genes encompassed within these particular datasets are as follows: Beroukhim: ALDOA, ALDOC, ENO2, GPI, HK2, LDHA, PGK1. Gumz: ADPGK, ALDOA, ENO2, HK1, HK2, LDHA, PFKP, PGAM1, PGK1, PKM2, TPI1. Yusenko: ALDOA, ALDOC, ENO1, ENO2, GAPDH, HK1, HK3, LDHA, PFKL, PFKP, PGAM1, PGK1, PKM2, TPI1.

(2) Regarding the enrichment of DEGs, we posit that the enrichment of DEGs in the glycolytic pathway arises from the partial overlap between some genes in the glycolysis-related gene set and the DEGs. By querying the relevant pathway on the KEGG official website, we obtained the KEGG_GLYCOLYSIS_GLUCONEOGENESIS dataset comprising 62 genes. Notably, this dataset is broader in scope compared to the more focused criteria we employed for screening. Additionally, our screening process incorporated three distinct glycolysis-related datasets, inherently resulting in a reduction in the final count of acquired genes. In essence, the reference datasets utilized in both the DEGs enrichment process and the subsequent glycolysis-related selection process are distinct.

To provide a more precise depiction of the ENO2 selection process, we have revised the description of the screening process in Result 3.3 as follows: 'We conducted an overlap analysis of the DEGs with three independent datasets of glycolytic core genes (Beroukhim, GUMZ, Yusenko), leading to the identification of ENO2.

  1. For all micrograghs, the scale bars are required (Fig.3D, Fig.5H). For immunoblots (Fig.3F, Fig.5A), the molecular markers should be shown.

The reviewer’ s point is well taken. We have incorporated scale bars in Fig. 3D, Fig. 5H, and Fig. 5I, along with specifying their respective lengths in the figure legends. Additionally, we have enhanced all Western blot bands in the manuscript with molecular weight markers, including those in Fig. 3F and Fig. 5A.

  1. In Figure 5B, the bar graghs should be shown as other ones (Figure 5C, D, E) with individual data points.

We thank the reviewer for the thoughtful comments. We have revised the data presentation format of the bar chart in Figure 5B to align with the consistent format used in Figures 5C, 5D, and 5E.

  1. In ccRCC, HIFs are activated. The mechanism of HIFs regulating their target genes (Yang et al, Nat Commun, 2022 Jan 14;13(1):316; Zhang et al., Science, 2018 Jul 20;361(6399):290-295.) and the relationship of ENO2 and HIFs (Leiherer et al., Mol Cell Endocrinol. 2014 Mar 5; 383(1-2): 21–31; Fang et al., Blood 2009 Jul 23;114(4):844-59.) should be mentioned either in the “Introduction” or “Discussion” section. But, it’s better to confirm this by experiments in one of the ccRCC cell lines. Is ENO2 a direct HIFs target gene? Does HIFs bind to ENO2 promoter?

We extend our utmost gratitude to the reviewer for providing pertinent literature, which has greatly enriched our study. Building upon these contributions, we have undertaken significant revisions. Firstly, we have reworked the Discussion section, with a particular emphasis on the fourth paragraph of the Discussion on page 18. This revision delves into the specific mechanisms through which HIFs regulate target genes and discusses the reported association between ENO2 and HIFs. The specific modifications made are as follows: In the landscape of ccRCC, the HIF pathway assumes paramount importance. Ac-tivated in response to VHL gene mutations frequently observed in ccRCC, HIF engages with the 5′-(A/G)CGTG-3′ hypoxia response elements (HREs) within target gene pro-moters, instigating their transcriptional activation (44-46). Hypoxia stands as a pre-dominant component of solid tumors, primarily arising from compromised angiogene-sis post rapid tumor expansion, thereby yielding pathophysiological microcirculatory perturbations. HIF1α assumes a critical role in orchestrating tumor metabolism to re-spond to the increasingly hypoxic microenvironment (47). Oxygen limitation, a known driver of glycolysis, necessitates reliance on glycolysis for ATP production due to lim-ited oxidative phosphorylation. HIF1α is pivotal in this process, inducing the expres-sion of glycolytic enzymes such as hexokinase and phosphofructokinase, thereby sus-taining ATP generation (48,49). Notably, existing research has indicated a positive regulatory relationship between HIF1α and ENO2. Fang et al. demonstrated that hy-poxia upregulates the expression of both HIF1α and ENO2 in human myeloid cells (50). In a related context, Leiherer et al. confirmed the interaction between HIF1α and HREs of enolase 2 (ENO2), a key enzyme in glycolytic metabolism, within adipocytes (30). These studies highlight the intricate interplay between HIF1α and ENO2 across diverse cellular contexts. Our investigation further underscores the HIF1α-mediated tran-scriptional regulation of ENO2 in ccRCC, emphasizing the pivotal role of HIF1α in metabolic control.

Secondly, within the Result 3.5, we have augmented the description of the experimental elucidation of the precise mechanism underlying HIF1A's regulation of ENO2. Our experimental data, encompassing Western blot and qRT-PCR analyses, substantiate that HIF1A knockdown leads to a downregulation of ENO2 expression. Furthermore, the implementation of ChIP assays and dual luciferase reporter gene experiments conclusively demonstrates that HIF1A binds to the hypoxia-responsive element (HRE) within the ENO2 promoter region, thereby activating ENO2 transcription.

  1. It would be better to validate the function of ENO2 in clear cell renal cell carcinoma in vivo animal model.

We are very grateful to the reviewer’ s careful comments of our manuscript. We indeed devised in vivo experiments to validate the impact of ENO2 on ccRCC, a facet which, due to considerations of manuscript length at the time, was not included in the initial submission. In light of the reviewer's constructive suggestion, we have incorporated the effects of ENO2 on subcutaneous tumor growth in nude mice in Result 3.6, as found on page 14 of the manuscript. Consistent with our in vitro findings, the knockdown of ENO2 significantly curtails subcutaneous tumor growth in nude mice.

  1. The writing of this manuscript need to be improved. For example, there are several typos, including but not limited to: (a) “…including HIF-2a and HIF-1a, are activated 0”; (b) Figure 5F, “time (days)”; (c) 5H, asterisks.

Thank you for your valuable feedback and for bringing these issues to our attention. We sincerely apologize for any typos that may have occurred in the initial submission. We appreciate your diligence in reviewing the revised manuscript and ensuring that these errors have been corrected. We have rectified the typos highlighted by the reviewer and conducted a thorough review of the manuscript's text, figures, and tables to ensure the complete elimination of any such errors.

Please carefully revise the manuscript. Many aspects of the writing of this manuscript need to be improved. For example, there are several typos, including but not limited to: (a) “…including HIF-2a and HIF-1a, are activated 0”; (b) Figure 5F, “time (days)”; (c) 5H, asterisks.

Thank you for your thorough review and constructive feedback. We appreciate your attention to detail and apologize for any typos present in the manuscript. We have taken your comments seriously and have diligently revised the manuscript to rectify these issues. Our aim is to ensure the highest quality of writing and presentation. Your input has been invaluable in improving the manuscript's accuracy and readability.

Reviewer 3 Report

In the present work, Shi et al. presented a hypothesis on the role of ENO2 as a biomarker for regulating energy metabolism in renal cell carcinoma, as well as a biomarker for tumor progression.

Their work is well written and presents their findings thoroughly. Their work is very interesting and it has merit for publication after addressing some minor issues.

Please comment on the role of ENO2 in energy metabolism. Since the authors mention the gene's role in glycolysis it would be interesting to further investigate the correlation of ENO2 expression and glucose consumption in cell line models. Please discuss this topic in the "Discussion" section.

The authors should highlight their findings and also mention how are their findings able to help to the diagnosis (prognosis ?) and therapy of renal cell cancer. Is it possible to use the gene/molecule as a therapeutic target?

Overall, this is a very interesting study and the authors have made a great deal of experimentation to support their hypothesis.

minor editing

Author Response

We are grateful for your time and effort in evaluating our manuscript and providing constructive suggestions to further enhance its quality.

1.Please comment on the role of ENO2 in energy metabolism. Since the authors mention the gene's role in glycolysis it would be interesting to further investigate the correlation of ENO2 expression and glucose consumption in cell line models. Please discuss this topic in the "Discussion" section.

We greatly appreciate the reviewer's feedback. Following the suggestions, we have augmented the third paragraph of the Discussion section on page 18 to include a comprehensive description of ENO2's role in energy metabolism and tumor progression. Furthermore, we have elaborated on our experimental findings concerning ENO2's involvement in regulating glucose metabolism in ccRCC cell lines.

Additionally, as per the reviewer's recommendation, we have conducted experiments detailed in Result 3.6 to elucidate ENO2's impact on ccRCC glucose metabolism. The outcomes of these experiments demonstrate that ENO2 promotes glycolysis in ccRCC cells.

2.The authors should highlight their findings and also mention how are their findings able to help to the diagnosis (prognosis ?) and therapy of renal cell cancer. Is it possible to use the gene/molecule as a therapeutic target?

We extend our sincere gratitude to the meticulous comments provided by the reviewer. We have undertaken a comprehensive revision of the Discussion section, emphasizing the potential implications of our results for the diagnosis and treatment of ccRCC. These modifications have been implemented across the entire Discussion section, spanning pages 17 to 19 of the manuscript. The revised "Discussion" section is as follows:

Here, we have elucidated ENO2 as a pivotal gene regulating the metabolic reprogramming of ccRCC through a series of bioinformatics analyses. ENO2 is upregulated in ccRCC and serves as a potential biomarker for this condition. Functional and mechanistic investigations have corroborated that ENO2 is transcriptionally activated by HIF1α, and it orchestrates tumor cell glycolysis to sustain the energy supply and promote the progression of ccRCC.

Previous investigations have identified numerous biomarkers for ccRCC. However, the clinical significance of most remains to be established. Through differential ex-pression analyses of multiple ccRCC gene expression datasets, we have revealed that in addition to its angiogenic role, glycolysis constitutes another pivotal factor influencing ccRCC initiation and progression. The Warburg effect stands as one of the earliest manifestations of metabolic reprogramming in ccRCC (31). Despite ample oxygen availability, a majority of cancer cells predominantly generate energy through glycolysis, in stark contrast to most normal cells that primarily utilize mitochondrial oxidative phosphorylation. This tumor-specific Warburg effect furnishes the energy and biosynthetic precursors requisite for propelling tumorigenesis (32,33). The Warburg effect exerts its influence across various facets of ccRCC tumorigenesis, encompassing tumor growth, survival, angiogenesis, and metastasis (34-36). Moreover, the acidic microenvironment resultant from lactate accumulation within the tumor milieu fosters invasive behavior and immune evasion (37,38).

ENO2 plays a pivotal role in energy metabolism and contributes to the metabolic reprogramming and progression of tumors (39). Enhanced glycolytic activity empowers cancer cells to rapidly generate energy and biosynthetic intermediates, thereby supporting their growth and survival. ENO2, by catalyzing a crucial step in glycolysis, promotes the conversion of 2-phosphoglycerate to phosphoenolpyruvate, thereby aiding in this metabolic adaptation (40). ENO2 has been upregulated in various tumor types, including head and neck cancer (39), colorectal cancer (41), pancreatic cancer (42), and bladder cancer (43), and has been associated with more aggressive phenotypes. Our investigations further confirm that heightened ENO2 expression correlates with elevated glucose uptake and lactate production, hallmark features of the War-burg effect. This observation underscores the role of ENO2 in sustaining the growth of tumors, including ccRCC. These findings shed light on the significance of ENO2 as a key regulator of tumor metabolism.

In the landscape of ccRCC, the HIF pathway assumes paramount importance. Activated in response to VHL gene mutations frequently observed in ccRCC, HIF engages with the 5′-(A/G)CGTG-3′ hypoxia response elements (HREs) within target gene promoters, instigating their transcriptional activation (44-46). Hypoxia stands as a pre-dominant component of solid tumors, primarily arising from compromised angiogenesis post rapid tumor expansion, thereby yielding pathophysiological microcirculatory perturbations. HIF1α assumes a critical role in orchestrating tumor metabolism to respond to the increasingly hypoxic microenvironment (47). Oxygen limitation, a known driver of glycolysis, necessitates reliance on glycolysis for ATP production due to limited oxidative phosphorylation. HIF1α is pivotal in this process, inducing the expression of glycolytic enzymes such as hexokinase and phosphofructokinase, thereby sustaining ATP generation (48,49). Notably, existing research has indicated a positive regulatory relationship between HIF1α and ENO2. Fang et al. demonstrated that hypoxia upregulates the expression of both HIF1α and ENO2 in human myeloid cells (50). In a related context, Leiherer et al. confirmed the interaction between HIF1α and HREs of enolase 2 (ENO2), a key enzyme in glycolytic metabolism, within adipocytes (30). These studies highlight the intricate interplay between HIF1α and ENO2 across diverse cellular contexts. Our investigation further underscores the HIF1α-mediated transcriptional regulation of ENO2 in ccRCC, emphasizing the pivotal role of HIF1α in metabolic control.

In conclusion, the research focused on the role of ENO2 in ccRCC has illuminated its pivotal involvement in the intricate landscape of tumor metabolism and progression. Through its participation in glycolysis, ENO2 emerges as a key player in fueling the energy demands of ccRCC cells, ultimately contributing to their rapid proliferation and survival. Furthermore, the identification of HIF1α-mediated transcriptional regulation of ENO2 adds a crucial layer to our understanding of the metabolic adaptations crucial for tumor growth in the context of ccRCC. The connection between ENO2 and the HIF pathway underscores the sophisticated interplay between metabolic reprogramming and the tumor microenvironment, highlighting the significance of ENO2 in translating the hypoxic response into altered glycolytic activity. As such, targeting ENO2 and its interactions with HIF1α could potentially offer novel therapeutic avenues for ccRCC treatment by exploiting the vulnerabilities that arise from metabolic adaptations. Overall, these findings underscore the importance of unraveling the molecular intricacies governing ccRCC metabolism, paving the way for potential advancements in both diagnosis and therapeutic strategies targeting this challenging malignancy.

Round 2

Reviewer 2 Report

The authors revised their manuscript and provided the relatively reasonable explanations to my questions. Some details were also added in this manuscript. This paper should be able to bring the readers some profound understanding about the role of ENO2 in clear cell renal cell carcinoma. I have no more comments for the current version.

Reviewer 3 Report

The authors have addressed all previous comments. Their manuscript can be published in its present form.